# Characterization and Pathogenicity of *Flavobacterium psychrophilum* Isolated from Rainbow Trout (*Oncorhynchus mykiss*) in Korea

**DOI:** 10.3390/microorganisms11102546

**Published:** 2023-10-12

**Authors:** Jiyeon Park, HyeongJin Roh, Yoonhang Lee, Ju-Yeop Lee, Hyo-Young Kang, Min Ji Seong, Yura Kang, Young Ung Heo, Bo Seong Kim, Chan-Il Park, Do-Hyung Kim

**Affiliations:** 1Department of Aquatic Life Medicine, College of Fisheries Science, Pukyong National University, Busan 48513, Republic of Korea; 2Department of Aquatic Life Medicine, College of Ocean Science and Technology, Kunsan National University, Gunsan 54150, Republic of Korea; 3Department of Marine Biology & Aquaculture, College of Marine Science, Gyeongsang National University, Tongyeong 53064, Republic of Korea

**Keywords:** *Flavobacterium psychrophilum*, rainbow trout, biochemical characteristics, genome analysis, pathogenicity

## Abstract

*Flavobacterium psychrophilum* is the causative agent of bacterial cold-water disease in salmonids and rainbow trout fry syndrome. This pathogen has attained a global presence and can spread both horizontally and vertically. However, it was not documented in Korea before September 2018. In this study, the objectives were to characterize *Flavobacterium psychrophilum* strain FPRT1, isolated from diseased rainbow trout genotypically and phenotypically. We also conducted various investigations to better understand its impact and assess potential control measures. We acquired fifty rainbow trout (approximately 70 g in weight) and transferred them to a laboratory aquarium. During the initial acclimation period, we observed mortality and examined affected fish for clinical signs. We isolated the bacterium from the spleen of infected rainbow trout using tryptone yeast extract salts agar supplemented with glucose, naming this FPRT1. Antibiotic susceptibility testing was carried out, and from the result, we selected enrofloxacin to administer to the trout orally to reduce mortality. To evaluate pathogenicity, we exposed the trout to FPRT1 at different water temperatures (8, 15, and 22 °C). Genomic analysis was conducted to identify the serotype and relatedness of FPRT1 to European strains. Affected fish displayed clinical signs, such as ulcerative lesions in the mandible, anemia with pale gills, exophthalmia, and increased mucus secretion. Internal symptoms included pale liver and enlarged spleen. FPRT1 was susceptible to erythromycin, enrofloxacin, florfenicol, oxytetracycline, and gentamicin, but resistant to oxolinic acid and sulfamethoxazole/trimethoprim. Oral administration of enrofloxacin resulted in a decrease in mortality from 28% to 6%. Pathogenicity tests revealed varying mortality rates due to FPRT1 at different temperatures. The highest rates were observed at 8 °C (ranging from 43% to 100%) for both intraperitoneal and intramuscular injections, and lower rates occurred at 22 °C (ranging from 0% to 30%), with intramuscular injections displaying higher susceptibility. Genomic analysis identified FPRT1 as serotype 2 and indicated its close genetic relationship with European strains based on the core genome and dispensable genome. The substantial genomic similarity between our strain and European strains suggests the possibility of bacterial spread through the importation of fertilized eggs from Europe. In conclusion, this study highlights the introduction of the previously undocumented pathogen (*F. psychrophilum*) into Korean rainbow trout populations. The detection of this pathogen and its pathogenicity assessment is not only important for understanding its impact on local aquaculture but also for establishing surveillance and control measures to prevent further transmission and outbreaks in the region.

## 1. Introduction

*Flavobacterium psychrophilum* is one of the most important pathogens in cool freshwater-farmed fish [1]. *F. psychrophilum* was first reported in North America in 1948 [2] and has since been detected in other countries, including France [3], Germany [4], Denmark [5], and Japan [6]. This pathogen has also been reported in outbreaks affecting a number of other fish species, including Atlantic salmon (*Salmo salar*) [7], coho salmon (*Oncorhynchus kisutch*) [2], and ayu (*Plecoglossus altivelis*) [8], and is well known as the causative agent of rainbow trout fry syndrome (RTFS) in rainbow trout (*Oncorhynchus mykiss*) and, more generally, of bacterial cold-water disease (BCWD) in many species [5]. *F. psychrophilum* not only infects fry fish, but also causes disease in adult fish, resulting in significant economic losses to salmonid fish farming worldwide [9,10,11,12]. The disease is characterized by anemia, external symptoms encompassing tail erosion, damage to tissues around the dorsal fin (saddleback lesion), and systemic spread to internal organs, such as the spleen and kidney [1,13,14]. Although the mechanisms behind the pathogenicity of *F. psychrophilum* in rainbow trout are not well understood, it is speculated that proteases and adhesion, motility, and a specific secretion system may be involved [15]. 

Rainbow trout ranks among the most commonly farmed fish worldwide, celebrated for its ease of reproduction, fast growth, and resilience to varying environments and diseases [16]. Looking at the domestic production of rainbow trout in South Korea over the last 5 years has shown a steady increase, rising from 2965 tons in 2015 to 3095 tons in 2019 [17]. This makes it the second most produced species in the country’s inland aquaculture industry, following eel, with a consistent annual production exceeding 3000 tons [18]. However, as production intensifies, rainbow trout exposed to such intensive conditions become more vulnerable to disease outbreaks caused by viral and bacterial pathogens, a concern shared across the aquaculture industry [19].

In Korea, the first reported cases of *F. psychrophilum* were isolated from ayu (*Plecoglossus altivelis*) in 1998 by Lee and Heo (1998) [20]. South Korea had been generally considered free of *F. psychrophilum* infection, as it had not been detected in surveillance programs across the country. However, in 2018, *F. psychrophilum* was isolated from diseased rainbow trout purchased from a fish farm in Korea. We confirmed that the isolate (FPRT1) was pathogenic as it caused mortality in rainbow trout. In this study, we have characterized the first *F. psychrophilum* isolate from rainbow trout in Korea in terms of pathogenicity, antibiotic susceptibility, and epidemiological features, and provide further valuable insights for disease management and prevention strategies in aquaculture.

## 2. Materials and Methods

### 2.1. Mortality in Rainbow Trout and Antibiotic Treatment

Fifty rainbow trout (body weights = ~70 g) were purchased from a fish farm in Korea, where the water temperature was maintained at approximately 18 °C. The fish were placed in a 500 L tank and acclimated at 15 °C. The fish were fed with commercial dry pellet (Sajo, Seoul, Republic of Korea) at a rate of up to 1.5% of their body weight per day. Within five days of arriving at the university’s aquarium, the fish began to die, and cumulative mortality was 28% by day 24. In order to treat the infection, the fish were fed pellets supplemented with enrofloxacin (Daehan New Pharm, Hwaseong, Republic of Korea) at a final concentration of 5 mg/kg/day for 8 days. The water temperature was also raised from 15 °C to 19 °C during the treatment period.

### 2.2. Histopathology

Moribund fish were euthanized using MS-222 (Sigma, St. Louis, MO, USA) before dissection and various organs were observed with routine histopathological techniques. Briefly, gills, stomach, liver, spleen, and kidneys were fixed in 10% neutral buffered formalin (BBC Biochemical, Washington, DC, USA) for 48 h, and then fixed samples were dehydrated and embedded in paraffin wax, sectioned at 4 μm, and stained with hematoxylin–eosin (H & E; BBC Biochemical, Washington, DC, USA) following routine protocols.

### 2.3. Bacterial Isolation and Identification

Moribund or dead fish were examined to identify the causative agent of disease. The liver, kidney, and spleen of dead fish were used to extract DNA using an AccuPrep^®^ Genomic DNA Extraction Kit (Bioneer, Daejeon, Republic of Korea). PCR was performed with the following primer pairs: 27F (5′-AGAGTTTGATCMTGGCTCAG-3′) and 1492R (5′-TACGGYTACCTTGTTACGACTT-3′) for 16S rRNA gene, and PSY1 (5′-GTTGGCATCAACACACT-3′), PSY2 (5′-CGATCCTACTTGCGTAG-3′) for *F. psychrophilum* specific 16s rRNA gene [6]. PCR product was loaded on a 1% (*w*/*v*) agarose gel containing ethidium bromide and electrophoresed for 20 min. The base sequences obtained via the Sanger sequencing method were blasted from NCBI (https://www.ncbi.nlm.nih.gov/nucleotide/, accessed on 18 September 2018). Bacteria were isolated from the liver, spleen, and kidney with pure culture on tryptone yeast extract salts agar supplemented with glucose (0.5 g L^−1^) (FLPA) according to Cepeda et al., 2004, which we named FPRT1 [21].

### 2.4. Biochemical Characteristics and Antibiotic Susceptibility Testing

API ZYM strips (bioMérieux, Marcy-L’Etoile, France) were used to test the activity of 19 enzymes. The experimental procedure followed the manufacturer’s instructions, and the results were read after incubation at 15–20 °C for 16–20 h. Disk diffusion assays were conducted following Clinical and Laboratory Standards Institute (CLSI) guidelines [22]. The assays were performed on modified Muller–Hinton agar (MHA) (Oxoid, Cambridge, UK) diluted at a ratio of 1:7, containing 5% fetal bovine serum (Gibco, Grand Island, NY, USA), at 15 °C for 72–96 h. The discs used were obtained from Oxoid and BD (BD Biosciences, Oxford, UK). The ten antibiotics used for the testing comprised: cefadroxil (CFR, 30 μg), enrofloxacin (ENR, 5 μg), oxolinic acid (OA, 2 μg), gentamycin (CN, 10 μg), erythromycin (E, 15 μg), oxytetracycline (OT, 30 μg), florfenicol (FFC, 30 μg), trimethoprim (W, 13 μg), sulfamethoxazole (RL, 25 μg), and trimethoprim/sulfamethoxazole (STX, 15 μg). The reference strains used were *A. salmonicida* ATCC 33658 and *E. coli* ATCC25922, which were incubated at 28 °C for 24 h.

### 2.5. Genome Comparison and Molecular Serotyping of FPRT1

Edgar 3.0 software [23] was used to analyze phylogenetic relationships based on core genes using 80 *F. psychrophilum* strains. This analysis updated the phylogenetic tree present in a previous study [24]. Multiple alignments of each orthologous gene set in the core genome were performed using MUSCLE version 3.2 [25], and resulting alignments were performed using the neighbor-joining method implemented in the PHYLIP package [26] to generate a phylogenetic tree. In addition, dispensable genomes from all 80 strains were utilized for dimensionality reduction and cluster division using uniform manifold approximation and projection (UMAP) and k-means analysis in R. The resulting analysis was visualized using RStudio 4.2.0 and R version 3.6.1. Serotypes were determined by in silico PCR amplification (http://insilico.ehu.es/PCR/, accessed on 10 March 2023) through a multiplex PCR-based serotyping assay developed by Rochat et al. (2017) [27].

### 2.6. Pathogenicity Tests

A total of 225 rainbow trout (average weight = 2.26 ± 0.71 g and length = 5.58 ± 0.56 cm) were purchased from a fish farm in Korea. Fish were acclimated in a 500 L tank and maintained at 15 °C for a week. Prior to commencing the artificial infection experiment, they underwent testing for bacteria, parasites, and viruses to ensure their disease-free status. Fish were fed with commercial dry pellets (Sajo, Seoul, Korea) at up to 2% of kg body weight/day. Before the experiment, the temperature was controlled to reach the target water temperature by gradually adjusting the water temperature at a rate of 1 °C per day for groups at both 8 °C and 22 °C. Among the standardization of experimental infection methods presented by Garcia et al., 2000, two methods were used: intraperitoneal and intramuscular injection [28]. Trial 1 involved the separation of fish (average weight = 2.26 ± 0.71 g and length = 5.58 ± 0.56 cm) into tanks (10 L, 10 fish per tank) and challenged them with FPRT1 at three different concentrations (10-fold serial dilutions: 8.44 × 10^7^ to 10^5^ CFU/mL) and three different temperatures (at 8, 15, 22 °C) via intraperitoneal (IP) injection of 0.1 ml fish^−1^. Controls were inoculated with PBS via the same route. After injection, mortality was monitored for 21 days, and 100% of the water in each tank was changed every 48 h. In Trial 2, fish (average weight = 2.7 ± 0.5 g and length = 6.1 ± 0.5 cm) were separated into tanks (10 L, 7 fish per tank) and challenged with FPRT1 at four different concentrations (10-fold serial dilutions: 4 × 10^6^ to 10^3^ CFU/mL by intramuscular (IM) injection with 0.05 mL fish^−1^) and three different temperatures (8, 15, 22 °C). Controls were inoculated with PBS via the same route. After injection, mortality was monitored for 21 days, and 100% of the water in each tank was changed every 48 h. *F. psychrophilum* was reisolated from dead fish and surviving fish were sacrificed on day 22. The 50% lethal dose (LD_50_) was calculated using the method of Reed and Muench [29]. This study was reviewed and approved by the Animal Research Ethical Committee at Pukyong National University (approval number: PKNUIACUC-2021-14).

## 3. Results and Discussion

### 3.1. Phenotypic and Molecular Identification

Most dead fish showed signs of disease, which included ulcerative lesions in the mandible, pale gills (anemia), exophthalmia, and/or increased mucus secretion. They also showed pale liver, enlarged spleen, and petechiae on the body surface (Figure 1A,B). This appears to be identical to one of the typical symptoms known in the case of *F. psychrophilum* infection, as indicated by anemia, resulting in pale gills, kidney, intestine, and liver [14]. Pure yellowish colonies were successfully isolated from the spleen, head kidney, and liver of two out of the seventeen diseased rainbow trout and identified as *F. psychrophilum* using 16s rRNA primer and PSY-1 and PSY-2.

### 3.2. Histopathology

Analysis revealed various histological changes in most organs of diseased rainbow trout. In particular, necrosis (pyknosis and karyolysis), a typical inflammatory response, was observed in the kidney (Figure 1C,D), gills (Figure 1E), liver (Figure 1F), and spleen (Figure 1G). The stomach exhibited histopathological abnormalities, showing vacuolation. (Figure 1H). One of the symptoms of *F. psychrophilum* infection, inappetence, is predicted to result from insufficient food intake, leading to the development of vacuolation [14]. 

### 3.3. Biochemical Characteristics of Isolate FPRT1

Based on the API ZYM testing, our isolate was found to respond positively to alkaline phosphatase, esterase (C4), esterase lipase (C8), leucine arylamidase, acid phosphatase, and naphthol-AS-BI-phosphohydrolase. It appears that *F. psychrophilum* produced lipolytic and proteolytic enzymes but did not produce enzymes involved in carbohydrate metabolism. By comparing enzyme activities using API ZYM among *F. psychrophilum* strains from different countries, we found differences in lipase and peptide hydrolases (cystine arylamidase, and trypsin) compared to strains from other countries. We speculate that our strain and the Danish strain [30] have the same enzyme activities (Appendix A).

### 3.4. Antibiotic Susceptibility

To date, there is no commercially available vaccine to treat this disease; therefore, antibiotics are widely used in its treatment [31,32]. The diameter of the inhibition zone for each antibiotic was as follows: CFR, 43 mm; ENR, 24 mm; CN, 22 mm; E, 25 mm; OT, 44 mm; FFC, 48 mm; and OA W, RL, and STX were 6 mm. When comparing the results with previously reported strains from Turkey [10], our strain was susceptible to E, ENR, FFC, OT, and CN, while being resistant to OA, W, RL, and STX, which showed a 6 mm inhibition zone. After oral administration of enrofloxacin, mortality decreased from 28% to 6% (Figure 2). There were no further deaths after 8 days. *F. psychrophilum* strains have been found to be susceptible to antibiotics such as FFC and ENR. Kum et al. (2008) reported that the *F. psychrophilum* showed 15% and 25% resistance to ENR and FFC, respectively [7]. In the minimum inhibitory concentration (MIC) results of *F. psychrophilum* isolates of Chilean origin, 38% resistance to ENR and 2% resistance to FFC were observed, indicating that both antibiotics could be used as appropriate therapeutic agents [33]. However, in 133 isolates from the UK, 85% of the MICs for ENR were classified as non-wild-type, showing regional differences and indicating that antibiotic susceptibility testing is essential before embarking upon antibiotic treatment [34].

### 3.5. Water-Temperature-Dependent Infections of Rainbow Trout

In the IP route challenge, the highest mortality rate of 80% occurred at 8 °C, and the LD_50_ value of FPRT1 was determined to be 8.44 × 10^6^ CFU/mL (Figure 3A). At 15 °C, mortality rates at 8.44 × 10^7^, ~10^6^, and ~10^5^ were 60%, 40%, and 30%, respectively, and the LD_50_ value of FPRT1 was determined to be 2.38 × 10^7^ CFU/mL (Figure 3B). At 22 °C, mortality rates at 8.44 × 10^7^, ~10^6^, and ~10^5^ were 30%, 0%, and 10%, respectively, and the LD_50_ value could not be derived (Figure 3C). Similarly, in the case of IM injection, the highest mortality rate was at 8 °C, with rates in the range 4.0 × 10^6–3^ CFU/mL of 100%, 86%, 86%, and 43%, respectively, and the LD_50_ value of FPRT1 was determined to be 3.61 × 10^3^ CFU/mL (Figure 3D). At 15 °C, mortality rates at doses of 4.0 × 10^6–3^ were 100%, 57%, 29%, and 14%, respectively, and the LD_50_ value of FPRT1 was determined to be 2.39 × 10^5^ CFU/mL (Figure 3E). At 22 °C, mortality rates at the same dose range were 29%, 29%, 14%, and 0%, respectively, and the LD_50_ value could not be determined (Figure 3F). At a concentration of 4 × 10^6^ CFU/mL, injection into the muscle resulted in a mortality rate that was about 3.6 times lower at the lowest temperature and 6.1 times lower at a concentration of 4 × 10^4^ CFU/mL, indicating the impact of temperature on pathogenicity. Rucker et al. (1953) conducted an experiment in which a group of juvenile coho salmon, afflicted by BCWD, were moved to a separate trough and subjected to an increased water temperature, transitioning from a range of 4–6 °C to 13 °C [35]. Mortality declined sharply after a few days among these fish, and no *F. psychrophilum* could be isolated after one week. Preventive measures against this disease are notably sensitive to water temperature, with reports indicating that the disease tends to manifest below 15 °C and is generally absent above 20 °C. This highlights the critical role of temperature in determining disease susceptibility [1,2]. Consistently, the results of our study also revealed a similar pattern, as the highest mortality rate was observed at 8 °C, reaffirming the significance of water temperature in *F. psychrophilum* infections. Understanding the infection pathways and implementing appropriate preventive measures are crucial for improving the welfare and production of salmonid fish and may lead to more effective aquaculture practices.

### 3.6. Molecular Serotyping

Based on the multiplex PCR method [19], our strain belongs to serotype 2, which corresponds to serotype Th according to the study by Lorenzen and Olesen (1997) and serovar 2b or 3 according to the study by Mata et al. (2002), all of which are exclusively isolated from rainbow trout [36,37]. Different serotyping methods have led to varying proposals regarding the number of serotypes among *F. psychrophilum* isolates. Early studies by Izumi and Wakabayashi (1999), and Izumi et al. (2003) divided a collection of Japanese isolates from coho salmon, ayu, rainbow trout, and amago (*O. masou rhodurus*) into four serotypes (O-1, O-2, O-3, and O-4) [38,39]. Lorenzen and Olesen (1997), focusing on isolates mainly from rainbow trout, identified one predominant serotype (serotype Th; subtypes Th-1 and Th-2) that represented the majority of Danish isolates and isolates from other European countries [36]. They also found two minor serotypes (Fd and Fp^T^). Serotype Fd consisted of only a few isolates, while serotype Fp^T^, defined by the type strain *F. psychrophilum* NCIMB 1947T, predominantly included isolates from asymptomatic fish or fish species other than rainbow trout. Mata et al. (2002) classified *F. psychrophilum* isolates into seven host-dependent serovars, with serovar 1 found only in salmon isolates, serovars 2 and 3 found exclusively in rainbow trout isolates, and the remaining serovars associated with specific fish species (4: eel; 5: carp; 6: tench; 7: ayu) [30]. Serovar 1 had previously been referred to as Fp^T^ [36] or O-1 [38], while serovars 2 and 3 were previously described as types Th and Fd [36] or O-3 [38]. The diverse serological characteristics of *F. psychrophilum* strains not only facilitate the selection of strains for vaccine development, but also contribute to epidemiological surveillance, disease control, and the understanding of virulence and host resistance [27,34,40].

### 3.7. Phylogenetic Clustering of F. psychrophilum Using Genomes

In recent years, the study of bacterial genomes has revolutionized the field of epidemiology, providing a powerful tool for understanding the transmission and spread of infectious diseases [41,42]. Since 2007, several studies have used the genome of *F. psychrophilum* to investigate its genetic diversification and virulence characteristics [13,15,30,43]. Based on phylogenetic analysis using 1187 genes, the core genomes of 80 *F. psychrophilum* strains derived from Europe, America, and Asian countries were divided into three main clusters (Appendix A). FPRT1 belongs to the first cluster and shows a close relationship to European strains, with all but one strain from Chile being of European origin. Cluster 2 is composed of fish-derived strains from Europe, America, and Asia, while Cluster 3 comprises rainbow-trout-derived strains from Europe and America.

UMAP is a dimensionality reduction technique that allows for the visualization of high-dimensional data in a lower-dimensional space [44]. It simplifies genomic data by identifying patterns and relationships among different genomic sequences, providing insights into genetic diversity and relationships between strains or populations [45]. Using UMAP analysis on genomic data from the same set of 80 strains used in the core genome-based phylogenetic analysis, the strains were divided into four clusters. FPRT1 belongs to Cluster 3, which consists solely of strains from European and rainbow trout origin (Figure 4). In contrast, Cluster 1 contains a variety of fish species originating in Europe, America, and Asia, while Clusters 2 and 3 are composed of rainbow-trout-derived strains from Europe and America. Notably, clusters showed clearer divisions based on country and fish species origin compared to the core genome-based phylogenetic tree.

The horizontal transmission and vertical infection of *F. psychrophilum* through fertilized eggs plays a crucial role in the bacterium’s transmission, facilitating its spread between continents via infected egg imports [9,46]. Kumagai and Takahashi (1997), for instance, detected *F. psychrophilum* in coho salmon eggs imported from the USA to Japan [47]. Similarly, MLST analysis of *F. psychrophilum* isolated from Chile revealed that the majority of strains were classified as CC-ST2, with predominant distributions in Europe, Japan, and North America [48]. In fact, rainbow trout in Korea are cultured through the importation of fertilized eggs, and recent imports have increased from 154 kg in 2012 to 457 kg in 2016 [49]. Most of the eggs in Korea are imported from European countries, and rainbow trout used in this study originally came from Denmark.

## 4. Conclusions

In this study, we have successfully identified the initial case of *F. psychrophilum* isolated from diseased rainbow trout in Korea. Through rigorous pathogenicity testing, we have confirmed that FPRT1 is responsible for causing disease, even leading to mortality. In addition, our analysis has revealed that the isolated strain bears a remarkably high similarity with European strains in both API ZYM tests and genomic analysis. This raises concerns about the potential transmission of this bacterium through the importation of fertilized eggs from Europe. Given the fastidious nature of *F. psychrophilum* and the absence of commercialized vaccines, it is imperative to establish stringent biosecurity measures within Korean fish farms to curtail the spread of this infection. Additionally, regular health checks for aquaculture facilities and the ongoing monitoring through the utilization of molecular diagnostic techniques can help reduce the introduction and spread of emerging diseases.

## Figures and Tables

**Figure 1 microorganisms-11-02546-f001:**
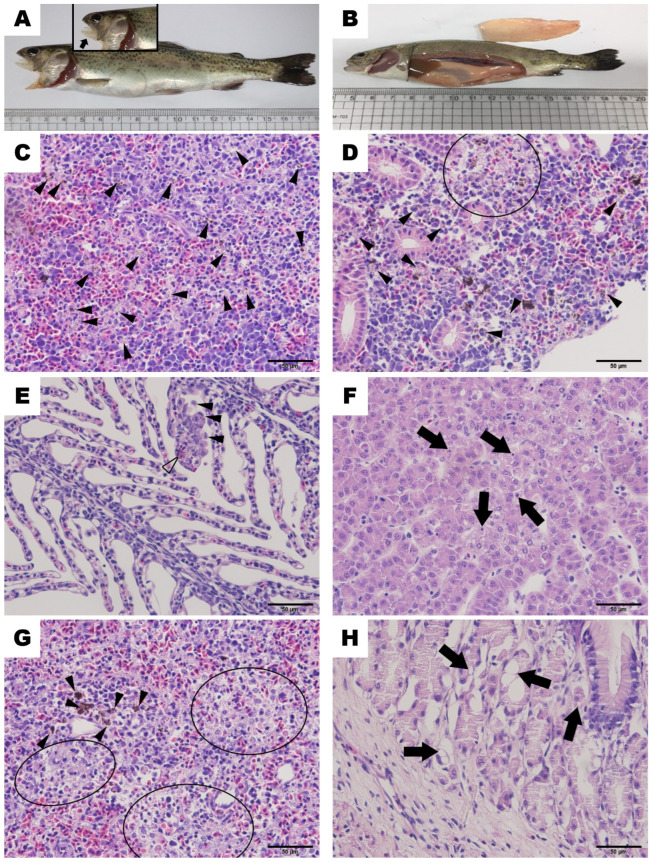
Clinical signs and histopathological observations of rainbow trout infected with *Flavobacterium psychrophilum* FPRT1 in this study. (**A**) Ulcerative lesions in mandible and extended abdomen. (**B**) Pale gills and liver congestion. (**C**) Numerous macrophages (arrowheads) containing melanin granules in the head kidney; (**D**) macrophages (arrowheads) containing melanin granules and macrophage aggregation (circle) in reticuloendothelial system of body kidney; (**E**) macrophages (transparent arrowhead) and monocyte infiltrations (arrows) in secondary lamella of gill; (**F**) disseminated coagulation necrosis with pyknosis (arrows) of hepatocytes; (**G**) macrophage aggregation (ellipse) in ellipsoid of spleen and macrophages (arrowhead) containing melanin granules in reticuloendothelial system; (**H**) vacuolation (arrows) of gastric epithelial cells in the stomach.

**Figure 2 microorganisms-11-02546-f002:**
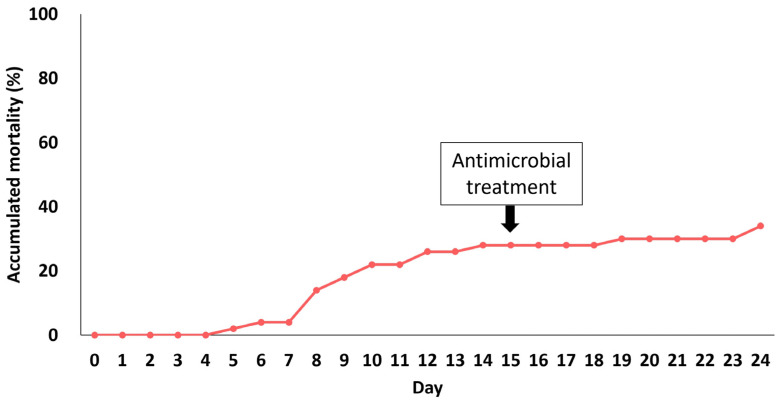
Cumulative mortality due to natural infection with *F. psychrophilum*. The cumulative mortality due to *F. psychrophilum* for 24 days is shown, and the arrow indicates the start date of antibiotic treatment.

**Figure 3 microorganisms-11-02546-f003:**
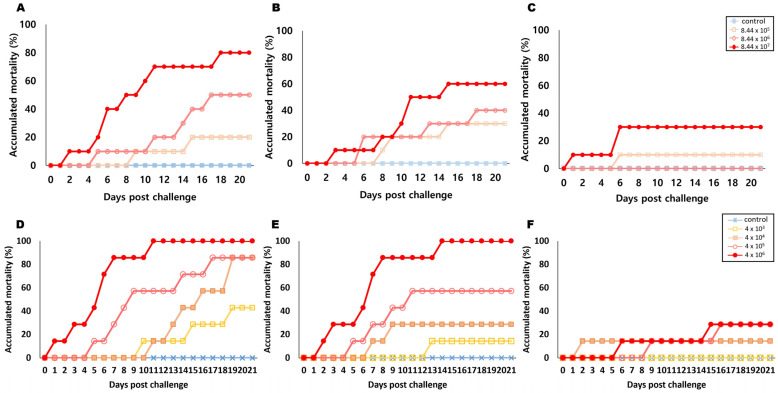
Cumulative mortality from *Flavobacterium psychrophilum* FPRT1 by intraperitoneal (IP) and intramuscular (IM) injection. Cumulative mortality from *F. psychrophilum* FPRT1 via IP injection is presented in (**A**–**C**), representing mortality rates at 8, 15, and 22 °C, respectively. Three different concentrations of *F. psychrophilum* were inoculated in the fish (10-fold serial dilutions: 8.44 × 10^5^ to 10^7^ CFU/mL). Blue (■), orange (□), pink (○), and red (●) represent the control, 8.44 × 10^5^, 8.44 × 10^6^, and 8.44 × 10^7^, respectively. Cumulative mortality from *F. psychrophilum* FPRT1 via IM injection is presented in (**D**–**F**), representing mortality rates at 8, 15, and 22 °C, respectively. Three different concentrations of *F. psychrophilum* were inoculated in the fish (10-fold serial dilutions: 4 × 10^3^ to 10^6^ CFU/mL). Blue(X), yellow (□), orange (■), pink (○), and red (●) represent the control, 4 × 10^3^ to 10^6^, respectively.

**Figure 4 microorganisms-11-02546-f004:**
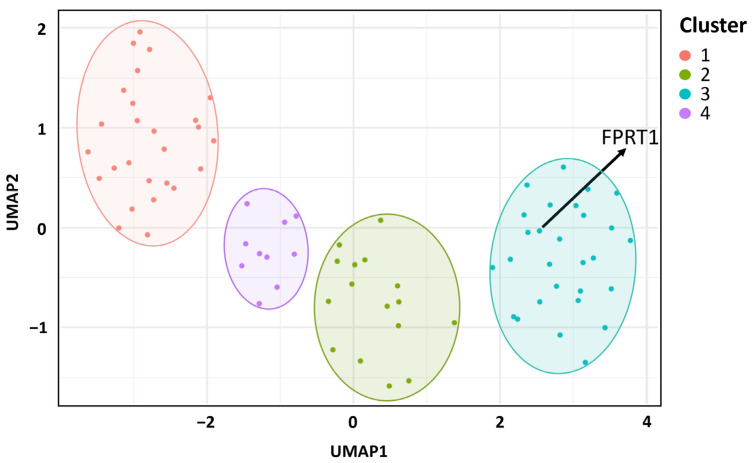
Uniform manifold approximation and projection (UMAP) clusters of 80 strains of *Flavobacterium psychrophilum* obtained from GenBank based on dispensable genome. Four individual clusters divided using 3045 genes are each shown in a different color. Cluster 1 contains strains isolated from salmonids (rainbow trout, coho salmon, Atlantic salmon, brown trout (*Salmo trutta*)) and various fish species in Europe, America, and Asia. Clusters 2 and 3 contain strains isolated from rainbow trout in Europe and America. Cluster 4 contains strains isolated only from European rainbow trout.

## Data Availability

The whole-genome sequences were deposited in GeneBank under the accession numbers CP059061.1.

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
