# Peer review of "Characterization and Pathogenicity of Flavobacterium psychrophilum Isolated from Rainbow Trout (Oncorhynchus mykiss) in Korea"

_microorganisms, 2023, doi:10.3390/microorganisms11102546_

Round 1

Reviewer 1 Report

Comments are listed in the attached file.

The manuscript's English expression needs improvement for clarity and precision

Author Response

First of all, thank you so much for your valuable comments. We have carefully reviewed the comments and have revised the manuscript accordingly. Our responses are given in a point-by-point manner in red color. We hope the revised version is now suitable for publication and look forward to hearing from you in due course.

Corrections are suggested below:

Abstract

  1. The abstract should clearly state the purpose of this study. It is advisable to structure the information into sections covering background, objectives, methods, results, and conclusions for enhanced readability.

Response: We appreciate your feedback. We attempted to re-organize the abstract into sections covering background, objectives, methods, results, and conclusions

  1. Line 37-39: It mentioned the potential bacterial spread through imported fertilized eggs, but it should emphasize the broader implications of the findings in the context of

fish farming, disease control etc.

Response: We appreciate the reviewer's feedback. We have added some sentences in lines 42-46 to address these important aspects and highlight the importance of our study for further contributions.

Introduction

  1. The introduction should be enriched with more background information, including the current status of the salmonid industry and the disease's impact on fish farming, underlining the study's necessity.

Response: Thank you very much for your suggestion. We have added a description in lines 87-97.

  1. Line 45: When starting a phrase, scientific names should be written in full. Check this in the whole text manuscript.

Response: Flavobacterium psychrophilum full name has already been specified in Line 72. And according to reviewer’s comment, we have checked the whole manuscript.

  1. Line 55: After providing the research background, you should explicitly mention your objectives or research questions, leading into your specific study.

Response: Thank you very much for your suggestion. We have already mentioned the purpose of the study in lines 103-106.

  1. Line 63: provide a brief description of the significance of this study.

Response: Thank you for your comments. We have incorporated the study's importance into lines 105-106.

Materials and Methods

  1. Line 75: How to anaesthetise fish before dissection? Please provide more details.

Response: Thank you for your comment. Before dissection, we were euthanized using MS-222. This information has also been included in line 121-122.

  1. Line 87: remove “by”

Response: According to reviewer’s comment, we have been removed “by”.

  1. Line 91: add brand and country manufactured for “tryptone–yeast extract–salts agar”.

Response: The medium used in ‘https://doi.org/10.1016/j.aquaculture.2004.05.013’ is not a commercially available product.

  1. Line 120-124: How long were the purchased fish temporarily reared, and were they tested for pathogens before the artificial infection experiment to ensure their health?

Response: The purchased fish were temporarily reared for approximately one week, and prior to the artificial infection experiment, we underwent testing for bacteria, parasites, and viruses to ensure their disease-free status.

  1. Line 127-133: Why were intraperitoneal and intramuscular injections chosen, and why were these concentration ranges of FPRT1 chosen for injection?

Response: Thank you for your comment. We have added information related to the infection route in lines 176-178. The concentration range for pathogenicity experiments was determined based on preliminary infection tests conducted at 15°C, using LD50 values obtained from those tests.

  1. Lines 133: to “103

Response: As per your comment, we have checked the manuscript and revised.

Results and discussion

  1. Line 142-144: Delete.

Response: As per your comment, we have removed line 195-197.

  1. Section 3.1: Please specify the number of fish evaluated and how many yielded positive isolation from tissues. Additionally, clarify from which organ the strain FPRT1 was isolated and the medium used for isolation.

Response: Thank you for your feedback. We have added the number of fish to line 205. Regarding the isolated tissues, it is mentioned in lines 204-205 as indicated that bacterial strain FPRT1 was isolated from the spleen, head kidney, and liver. Additionally, the method of using FLPA medium in the isolation process is clearly documented in lines 138-140 of the Materials and Methods section.

  1. Figure 1A does not clearly show ulcerative lesions on the mandible and abdomen, and Figure 1B exhibits atypical symptoms. I recommend replacing the symptom images with more representative ones.

Response: We appreciate the reviewer’s comment. We have made revisions to Figure 1A as suggested, while retaining Figure 1B as pale internal organs are also one of the symptoms.

  1. Lines 153-163: To ensure that the histological lesions are the result of a pathogenic infection, it is advisable to include histopathological images of healthy fish as controls.

Response: Thank you for your comment. During the study, histopathological examinations were exclusively performed on moribund individuals, and we did not possess examination results for healthy fish. In fact, we have examined healthy fish from the same farm we acquired for this study, and we did not detect any histopathological changes.

  1. Line 166-167: Place the corresponding figure number after the pathological description, it helps in better identifying the pathological changes in the study.

Response: Thank you for your feedback. We have revised line 223-226.

  1. Line 168: What histopathological abnormalities? please provide a brief description.

Response: Thank you very much for your valuable comment, the manuscript was revised as per the reviewer’s comment in L224-228.

  1. Section3.1-3.3: These sections of the results should be discussed appropriately, comparing and analyzing them in relation to findings from the literature.

Response: Thank you for the valuable comments. In accordance with the reviewer's feedback, we have included additional information in lines 202-204 and 226-228, comparing our findings with results from the literature.

  1. Line 181 and 185: change “inhibitor zone” to “inhibition zone”.

Response: We have made the requested correction by changing "inhibitor zone" to "inhibition zone" in lines 241 and 246.

  1. Line 189: add “%” after 15

Response: We have added "%" after "15" in line 249.

  1. Figure 2: Was there a control group that did not receive antibiotic treatment?

Response: Figure 2 indeed lacks a control group that did not receive antibiotic treatment. The reason for this absence is due to fish brought from the farm were acclimatized to one tank and disease occurred in that tank, so healthy fish could not be used as controls.

  1. Line 245-259: Emphasize the presentation of obtained results, followed by comparisons and discussions in the context of the data available in the literature.

Response: Thank you for your comment. We tried to re-organized the relevant result section according to your suggestions (line304-307).

  1. Line 253: FpT

Response: We have made the correction for " FpT " in line 315.

  1. Line 260-263: The serotype of the isolated strains should be visually represented through a phylogenetic tree.

Response: We appreciate your comment. The paper 'doi:10.3389/fmicb.2017.01752' already contains a comprehensive study on serotypes and their evolutionary relationships. Given the extensive information available in that source, we believe that creating an additional phylogenetic tree for serotypes in our paper may not be necessary.

  1. Line 288-289: explain the advantages of UMAP analysis compared to phylogenetic analysis and its significance for epidemiological and pathogen transmission research.

Response: Thank you for your comment. The analysis using UMAP has the advantage of identifying patterns and relationships among various genomes, simplifying the analysis of diversity and relationships between lineages. We have specified this in lines 342-345. Also, as mentioned in lines 351-351 UMAP-based classification clearly demonstrates the differentiation by fish species and country of origin. These results provide evidence that our strain is similar to European strains. If such analyses are applied in similar studies in the future, they could serve as valuable tools for epidemiological and pathogen transmission research.

  1. Line 302-311: Integrate the results in this section with findings from the literature for discussion.

Response: First, we have listed the results of our study as mentioned in lines 347-351. Subsequently, we referred to similar studies conducted on F. psychrophilum (lines 353-359). We then discussed that the FPRT1 strain isolated in our study is similar to European strains and suggested the possibility of bacterial spread through imports.

Conclusion

  1. Line 317-322: Consider highlighting the practical implications of the findings. What measures can be taken to mitigate the risk of introducing pathogens in the future? Please add it.

Response: Thank you for your comment. As per your suggestion, we have added a sentence (line 394-397) about the monitoring and surveillance of diseases.

Reviewer 2 Report

The manuscript titled "Characterization and pathogenicity of Flavobacterium psychrophilum isolated from rainbow trout (Oncorhynchus mykkis) in Korea" is original research article. The topic of the article is very interesting. Flavobacterium infection of fish is still an actual issue in salmonid farms worldwide. The authors have worked with results in many analyses involving both histology and genetics. I recommend the paper for publication after minor modifications. 

Abstract:

no comments

Introduction:

I recommned to add some more general informations about this severe disease.

L47 - divide the citations - France (3), Germany (4) etc.

L48-49 - also divide the citation

Materials and Methods:

L67 - add the information about the water temperature on the farm where the trouts came from (to know the difference of the temperatures)

L89 and 117 - the web page is not in references

L92 - I can´t find Cepeda in references

Results and discussion:

Figure 4 - cluster 1 contains strains from salmonids and .... from Europe, America and Asia

cluster 2 and 3 contains strains from O. mykiss from Europe and America - but this fish is aslo salmonid - so specify more the salmonid fish used for cluster one or write that you made special cluster for rainbow trout without using data from Asia??? I don´t understand it so I recommend to specify it

COnclusion:

no comments

References:

1/ please, go slowly through the list of references once more. There are mistakes as on L363 or 388 - the year of the publication is at the end of the reference not after the authors names

2/ in the whole long list of citations, not a single latin name of bacteria or fish is in Italics - and they should be

Author Response

The manuscript titled "Characterization and pathogenicity of Flavobacterium psychrophilum isolated from rainbow trout (Oncorhynchus mykkis) in Korea" is original research article. The topic of the article is very interesting. Flavobacterium infection of fish is still an actual issue in salmonid farms worldwide. The authors have worked with results in many analyses involving both histology and genetics. I recommend the paper for publication after minor modifications.

Thank you very much for sending the review’s valuable comments that were helpful in strengthening the quality of the manuscript. We have carefully reviewed the comments and have revised the manuscript accordingly. Our responses are given in a point-by-point manner in red color. We hope the revised version is now suitable for publication and look forward to hearing from you in due course.

Introduction:

  1. I recommned to add some more general informations about this severe disease.

Response: Thank you for the valuable comments. Additional information on symptoms and infection routes has been written on lines 81-84.

  1. L47 - divide the citations - France (3), Germany (4) etc.

Response: According to reviewer’s comment, we have divided the references line74-75.

  1. L48-49 - also divide the citation

Response: In response to the reviewer's feedback, we have split the citation in lines 76-77.

Materials and Methods:

  1. L67 - add the information about the water temperature on the farm where the trouts came from (to know the difference of the temperatures)

Response: As per your comment, we have added the information about water temperature in lines 114.

  1. L89 and 117 - the web page is not in references

Response: The journal's guidelines state, "Content that does not fulfill these criteria may be listed directly in the main text and might include company websites or websites to track project development." In line with this guidance and referencing a recent paper published in the Microorganisms journal (https://doi.org/10.3390/microorganisms11102411), we have made the following modifications in lines 140 and 168.

  1. L92 - I can´t find Cepeda in references

Response: According to the reviewer’s comment, we have added reference.

Results and discussion:

  1. Figure 4 - cluster 1 contains strains from salmonids and .... from Europe, America and Asia cluster 2 and 3 contains strains from O. mykiss from Europe and America - but this fish is aslo salmonid - so specify more the salmonid fish used for cluster one or write that you made special cluster for rainbow trout without using data from Asia??? I don´t understand it so I recommend to specify it

Response: Thank you for your feedback. In response to the reviewer's comment, we have included additional information about 'cluster1' in lines 374-375

References:

  1. please, go slowly through the list of references once more. There are mistakes as on L363 or 388 - the year of the publication is at the end of the reference not after the authors names in the whole long list of citations, not a single latin name of bacteria or fish is in Italics - and they should be

Response: Thank you for your comment. According to reviewer’s comment, we have checked the whole manuscript and reference.